# Candidate Regulatory Genes for Hindlimb Development in the Embryos of the Chinese Alligator (*Alligator sinensis*)

**DOI:** 10.3390/ani13193126

**Published:** 2023-10-06

**Authors:** Liuyang Yang, Mengqin Liu, Yunzhen Zhu, Yanan Li, Tao Pan, En Li, Xiaobing Wu

**Affiliations:** 1College of Life Sciences, Anhui Normal University, Wuhu 241000, China; yly950466@163.com (L.Y.); 17856872220@163.com (M.L.); zyz2059368613@163.com (Y.Z.); 18437973926@163.com (Y.L.); pantao@ahnu.edu.cn (T.P.); 2Anhui Provincial Key Laboratory of Conservation and Exploitation of Biological Resources, Anhui Normal University, Wuhu 241000, China

**Keywords:** Chinese alligator, differentially expressed genes, embryonic period, hindlimb development, transcriptome

## Abstract

**Simple Summary:**

A better understanding of the hindlimb developmental process will provide data support for the comparative evolutionary and functional genomics of crocodilians. For the Chinese alligator, the hindlimb is one of the main sources of power, and its development and differentiation will directly influence the survival ability in the wild. But little is known about the molecular mechanisms regulating the development and differentiation of the hindlimb of the Chinese alligator. In this study, RNA-sequencing technology was used to analyze the gene expression of the hindlimb in the Chinese alligator embryos at different stages (on days 29, 35, 41, and 46). Differentially expressed genes (DEGs) at different stages were identified. Our results will provide data support for the embryonic development process of the Chinese alligator.

**Abstract:**

Crocodilians, which are a kind of animal secondary adaptation to an aquatic environment, their hindlimb can provide the power needed to engage in various life activities, even in low-oxygen water environments. The development of limbs is an important aspect of animal growth and development, as it is closely linked to body movement, support, heat production, and other critical functions. For the Chinese alligator, the hindlimb is one of the main sources of power, and its development and differentiation will directly influence the survival ability in the wild. Furthermore, a better understanding of the hindlimb developmental process will provide data support for the comparative evolutionary and functional genomics of crocodilians. In this study, the expression levels of genes related to hindlimb development in the Chinese alligator embryos during fetal development (on days 29, 35, 41, and 46) were investigated through transcriptome analysis. A total of 1675 differentially expressed genes (DEGs) at different stages were identified by using limma software. These DEGs were then analyzed using weighted correlation network analysis (WGCNA), and 4 gene expression modules and 20 hub genes were identified that were associated with the development of hindlimbs in the Chinese alligator at different periods. The results of GO enrichment and hub gene expression showed that the hindlimb development of the Chinese alligator embryos involves the development of the embryonic structure, nervous system, and hindlimb muscle in the early stage (H29) and the development of metabolic capacity occurs in the later stage (H46). Additionally, the enrichment results showed that the AMPK signaling pathway, calcium signaling pathway, HIF-1 signaling pathway, and neuroactive ligand–receptor interaction are involved in the development of the hindlimb of the Chinese alligator. Among these, the HIF-1 signaling pathway and neuroactive ligand–receptor interaction may be related to the adaptation of Chinese alligators to low-oxygen environments. Additionally, five DEGs (*CAV1*, *IRS2*, *LDHA*, *LDB3*, and *MYL3*) were randomly selected for qRT-PCR to verify the transcriptome results. It is expected that further research on these genes will help us to better understand the process of embryonic hindlimb development in the Chinese alligator.

## 1. Introduction

The growth and development of animals are mainly controlled by key genes, which are significantly differentially expressed in cells at different developmental stages or in different tissues [1]; this is known as the spatiotemporal expression network of relevant regulatory genes in the field of developmental biology [2,3]. In the development of vertebrates, limb differentiation is an important step. The muscle growth and development process accompanying limb differentiation is also important, as it directly affects an individual’s ability to survive and move [4,5]. Muscles can be categorized into skeletal muscle, cardiac muscle, and smooth muscle, depending on their functional and morphological characteristics [6,7]. The growth and development of muscle tissue are processes involving protein accumulation, as well as cell proliferation and differentiation [8], including the formation of embryonic muscle fibers, the development of muscle fibers after birth, and the regeneration of adult muscle [9]. It has been shown that the number of muscle fibers in an animal is determined during the embryonic stage [10]; therefore, research on muscle development should comprehensively consider the early embryo.

Existing research on hindlimb development has mainly been carried out on model organisms and mammals; for example, acid signaling, FGF signaling, and *Meis* genes can regulate the initiation and formation of anterior–posterior patterns in mice [11,12]. Furthermore, muscle growth is a crucial component of limb development. The muscular system of fruit flies is mainly differentiated from the mesoderm, which, in its embryo, is subdivided into segmental units. The diversification of these units is influenced by signaling pathways, such as TGF-β and Wnt [13,14]. The myogenic regulatory factor (MRF) gene family mainly regulates the formation and development of porcine embryonic muscle fibers. In addition [15], *PITX1* is directly involved in the transcription of key components in the regulatory network of chondrogenesis and muscle formation in reptiles and mammals [16,17]. Muscle development and differentiation are the basis of a species’ ability to adapt to its environment and survive [18,19]. However, muscle development and differentiation involve many biological processes, and a large number of regulatory factors remain to be explored.

The Chinese alligator (*Alligator sinensis*) belongs to the family Crocodylidae and is a critically endangered crocodilian [20]. In order to adapt to its land movement, burrowing, and nesting functions, the hindlimb of the Chinese alligator is particularly developed compared with muscles in other areas of the body. The limbs of the Chinese alligator are short and powerful, consisting of a pair of forelimbs and a pair of hindlimbs, which present obvious differences; the forelimbs have five fingers and no webbing between the fingers, while the hindlimbs have four toes with webbing [21]. These structural characteristics represent adaptations of the Chinese alligator to its amphibious life. In addition, as a kind of secondary adaptation to an aquatic environment, the muscles of the Chinese alligator can provide the power needed to engage in various life activities, even in low-oxygen water environments. The tissue cell structure and function of the hindlimb of the Chinese alligator have been studied to a certain extent [22,23]; however, research on the hindlimb development of this animal remains relatively insufficient. The development of the hindlimb directly affects the development of the embryo and the physiological functions of the young crocodiles after breaking through and emerging from their shell, such as movement and feeding [24,25]. Therefore, it is of great theoretical and practical significance to study the hindlimb development of the Chinese alligator, as well as explore the genes involved in this process and the related regulatory mechanisms.

For this study, RNA-sequencing technology was utilized to investigate the hindlimb of the Chinese alligator embryos on days 29, 35, 41, and 46 of incubation (denoted by H29, H35, H41, and H46, respectively), aiming to characterize gene expression profiles during hindlimb development and growth in the Chinese alligator embryo and to identify key genes involved in this biological process. On this basis, the expression of genes related to the embryonic hindlimb development of the Chinese alligator is preliminarily discussed, providing data support for the comparative evolutionary and functional genomics of crocodilians.

## 2. Materials and Methods

### 2.1. Animals and Sample Preparation

For this study, the hindlimb of the Chinese alligator in the considered embryonic development stages was selected as the research object. The eggs considered in this study were produced from June to mid-July of 2020, and the sampling site was the Anhui Chinese Alligator National Nature Reserve, Xuancheng, Anhui, China. The incubation conditions of the fertilized eggs collected in the field are as follows: 32 °C with an ambient humidity greater than 90%.

According to the stages of embryonic development and the morphologic characteristics of appendages of the Chinese alligator [26,27], sampling was conducted after incubation of the Chinese alligator fertilized eggs for 29 days (H29), 35 days (H35), 41 days (H41), or 46 days (H46). Figure 1a–d represent the morphologic characteristics in H29, H35, H41, and H46, respectively. Additionally, the morphological characteristics of the Chinese alligator in each period are detailed in Table 1. Four biological replicates were taken from each period, and a total of 16 individuals were used for the experimental study. Fresh hindlimb tissue was cut into small pieces, immediately placed in RNA preservation solution, overnighted at 4 °C, and stored at −20 °C.

### 2.2. RNA Isolation, cDNA Library Construction, and Illumina Deep Sequencing

Total RNA was used as the input material for the RNA sample preparations. Total RNA was extracted using Trizol reagent (Invitrogen, Carlsbad, CA, USA), following the manufacturer’s protocol, and residual DNA was removed using DNase I (Promega, Madison, WI, USA). The mRNAs were enriched using magnetic Oligo(dT) beads for fragmentation. The first strand of cDNA was synthesized with random primers using the mRNA as a template. Second-strand cDNA synthesis was then performed to form a stable double-stranded structure; terminal repair was carried out, and a base was added to the 3’ end. Suitable fragments were purified with 2% agarose electrophoresis and then amplified and enriched via PCR, allowing the establishment of a cDNA library. A total of 16 cDNA effective libraries (concentration > 2 nM) were constructed with the hindlimb tissues. The cDNA library was sequenced using Illumina Hiseq 4000 (150 bp paired-end reads).

### 2.3. Assembly and Functional Annotation

Raw data (raw reads) in fastq format were first processed using in-house Perl scripts. In this step, clean reads were obtained by removing those containing adapters or poly-N, as well as low-quality reads (Q20 < 90%) from the raw data. All downstream analyses were conducted on the high-quality clean data.

Reference genome (GCF_000455745.1) and gene model annotation files were directly downloaded from the National Center of Biotechnology Information (https://www.ncbi.nlm.nih.gov/genome/?term=Chinese+alligator, accessed on 18 December 2021). The index of the reference genome was built using Bowtie v2.2.3, and clean reads were aligned to the reference genome using Trinity (https://informatics.fas.harvard.edu/best-practices-for-de-novo-transcriptome-assembly-withtrinity.html, accessed on 29 June 2023) [28]. Trinity has both reference-based and de novo processing features. This makes it all the more important that our study include the details of fundamentally important methods and statistical summaries of their libraries, filtering, mapping, assembly, and QC metrics, thereby including a presentation of the original data that were generated by this research. The genome mapping results are shown in Appendix A.

Subsequently, the fragments per kilobase of transcript per million mapped reads (FPKM) of each gene were calculated based on the length of the gene and read count mapped to the gene [29].

### 2.4. Identification of Differentially Expressed Genes

After data assembly, the fragments per kilobase of transcript per million mapped (FPKM) reads were used to calculate gene expression [29]. Pearson’s squared correlation coefficient (*R^2^*) and principal component analysis (PCA) were applied to better understand the repeatability across samples. Additionally, after filtering out the genes with FPKM = 0 and normalizing the FPKM via log2 transformation, the time sequence profile of gene expression was acquired.

The voom function in the limma package of R statistical software was applied to obtain the differentially expressed genes (DEGs). Significant DEGs were selected with multiple check corrections according to the standard criteria of *p*_adj_ < 0.01 and |log_2_(fold change)| ≥ 2.

### 2.5. Co–Expression Network Analysis and Visualisation

The R package (WGCNA) was used for weighted co-expression network analysis of the selected DEGs. Additionally, the automatic network building function blockwiseModules was used to detect the co-expression modules. The dynamic tree-cut algorithm was used to hierarchically cluster the identified modules. To explore the characteristic modules related to the development of the hindlimbs of the Chinese alligator embryos, the characteristic modules were associated with four developmental stages. The modules with greater correlation (|r| > 0.75, *p* < 0.05) were considered to be significantly related to the development of the hindlimb of the Chinese alligator. The top 5 genes for gene connectivity in each significantly correlated module were considered to be the key genes in the development of the embryonic hindlimbs of the Chinese alligator. Finally, each important module was visualized using Cytoscape software (version 3.10.0).

### 2.6. Functional Enrichment Analysis

Gene sets, such as the GO (Gene Ontology) and KEGG (Kyoto Encyclopedia of Genes and Genomes) databases, were selected to explain the biological functions of significant modules. The GO biological process items and KEGG pathways with *p* < 0.05 were considered to be significantly enriched.

### 2.7. Expression Analysis via Quantitative Real-Time PCR (qRT-PCR)

The β-actin gene of the Chinese alligator [30] was selected as the internal reference gene. Primer Premier 6.0 software was used to design qRT-qPCR primers. The qRT-PCR reaction system (10 μL) consisted of 5 μL of 2× SYBR qPCR Mix, 0.4 μL of forward and reverse primers, 0.1 μL of cDNA, and 4.1 μL of RNase-free H_2_O. The PCR program was implemented with the following incubation protocol: 15 min at 95 °C, 40 cycles lasting 10 s at 95 °C, 10 s at the annealing temperature (Appendix A), and 30 s at 72 °C, followed by the standard dissociation cycle. In order to detect any possible contamination, each reaction plate included no-template and no-reverse transcriptase controls. Each sample was analyzed in triplicate, and the relative levels of gene expression across samples were measured using the 2^−ΔΔCt^ method [31].

## 3. Results

### 3.1. Sequencing and Assembly Analyses

Transcriptome sequencing was performed on the hindlimb of the Chinese alligators. For this, four biological replicates were set in each period, and the 29-day-olds were used as the control group. A total of 713,362,246 original reads were obtained. After data filtering, reassembly, and removal of redundant data, 688,620,840 clean reads were obtained. The final data obtained for each sample were above 5.89 Gb. The GC content was between 49.25% and 50.8%, and the Q20 ratio was 97.3–97.7%, indicating that the sequencing and assembly data were of good quality and could be reliably used in the subsequent analysis. Please refer to Table 2 for specific data quality control results.

### 3.2. Analysis of Differentially Expressed Genes (DEGs)

The PCA results present changes in gene expression patterns from time to time or during development (Figure 2A).

After collating the transcriptome data, a total of 1675 genes were identified as DEGs over the four periods. The H29 vs. H46 control group had the largest number of DEGs (1493), among which 1028 were upregulated and 465 were downregulated. Additionally, the H41 vs. H46 control group had no DEGs.

The gene expression at H29 was compared with that at H35, H41, and H46, respectively. There were two downregulated DEGs between H35 and H29. There were 441 differentially expressed genes between H41 and H29, among which 216 were upregulated and 225 were downregulated. Lastly, a total of 1493 differentially expressed genes were detected between H46 and H29, with 1028 upregulated and 465 downregulated genes. As can be seen from Figure 2, there were significant differences (*p* < 0.05) in gene expression levels across the different periods of embryonic development for the Chinese alligator, among which the number of differentially expressed genes was the largest at H46 when compared with H29.

### 3.3. Co–Expression Network Analysis and Visualization

To evaluate the relationship between DEGs and hindlimb growth and development, we conducted a co-expression network analysis of all 1675 DEGs across four periods. The results show that WGCNA divides all DEGs into 7 modules with a soft threshold of 18 (Figure 3b,c). Among the seven modules, the turquoise module was identified as being significantly correlated with H29. In addition, the blue module, green module, and grey module were identified as being significantly correlated with H46.

### 3.4. Visualization of Significant Modules and Key Genes

To enhance our understanding of the DEGs involved in the hindlimb development of the Chinese alligator embryos, Cytoscape software was used to visualize the interaction networks of DEGs in significant modules. Additionally, we selected the top 50 DEGs in terms of gene connectivity with a weight greater than 0.15 in each important module to construct a gene co-expression network. After screening, 50, 34, 40, and 50 DEGs from the blue, green, grey, and turquoise modules were used to construct the gene co-expression network (Figure 4).

Based on the connectivity of DEGs, we screened the top five genes in each important module, which were considered core genes. A total of 20 hub genes were screened. For the expression of these hub genes, please see Appendix A. Among them, the turquoise module was significantly positively correlated with H29, suggesting that the turquoise module is involved in the early development of the hindlimb in the Chinese alligator. The blue, green, and grey modules were significantly positively correlated with H41 and H46, suggesting that these two modules play a positive role in the later developmental stage of the hindlimb of the Chinese alligator. In addition, there was a significant negative correlation between the grey module and H46, suggesting that the grey module plays a negative regulatory role in the later development of the hindlimb in the Chinese alligator.

### 3.5. Functional Enrichment Analysis

To study the function of four important modules in the development of the hindlimb of the Chinese alligator and the molecular mechanism by which DEGs regulate hindlimb development, GO and KEGG pathway enrichment analyses were carried out on the DEGs in the blue, green, and turquoise networks. The GO enrichment results showed that the DEGs in the blue modules were enriched with 55 biological processes, such as connective tissue development, cartilage development, and axon development (Figure 5a). DEGs in the green modules were enriched with 89 biological processes. DEGs in the grey modules were enriched with 28 biological processes, such as activation of an immune response, regulation of the immune effector process, and positive regulation of cell projection organization (Figure 5a). DEGs in the turquoise modules were enriched with 61 biological processes, such as gliogenesis, adult behavior, and locomotory behavior (Figure 5a). The significantly enriched GO molecular function and cellular component terms are listed in Appendix A. The KEGG enrichment results show that the DEGs in the blue module were significantly enriched in signaling pathways, such as the calcium signaling pathway, adipocytokine signaling pathway, and glycolysis/gluconeogenesis (Figure 5b). DEGs in the green module were significantly enriched in signaling pathways, such as cell adhesion molecules, inositol phosphate metabolism, and the notch signaling pathway. DEGs in the grey module were significantly enriched in signaling pathways, such as platelet activation, pantothenate and CoA biosynthesis, and beta-alanine metabolism. Additionally, DEGs in the turquoise module were significantly enriched in signaling pathways, such as neuroactive ligand–receptor interaction, the cell cycle, and motor proteins.

### 3.6. Validation of RNA-Seq via qRT-PCR

In order to verify the accuracy of the RNA sequencing, five differentially expressed mRNAs were randomly selected for quantitative verification via RT-qPCR. The corresponding differentially expressed genes were *CAV1*, *IRS2*, *LDHA*, *LDB3*, and *MYL3*, and the *β*-actin gene of the Chinese alligator was used as the internal reference gene. As shown in Figure 6, the fluorescence quantitative results demonstrate that the expression patterns of these five genes at the four embryonic stages were identical to the RNA-sequencing results. The expression trends were synchronized, and the results are consistent, proving the reliability of the RNA-sequencing results (in Figure 6, ** *p* < 0.01, and *** *p* < 0.001).

## 4. Discussion

Vertebrate limbs are a universal model for the study of the cellular and molecular interactions that determine morphological patterns during development [32]. Limbs have to meet the various survival needs of individuals, so they come in a variety of forms and structures [33]. Despite the differences in the adult muscle morphology among various reptile species, muscle development is highly conserved. Limb muscles are mainly derived from the paraxial mesoderm, which differentiates to form somites during development [34,35]. The dorsal somites develop into the myotome, which further develops into the skeletal muscles of the body and limbs [36]. During skeletal muscle development in vertebrates, the myotome undergoes sequential cell proliferation and differentiation into myoblast precursor cells, then myoblasts, and finally myocytes [37,38]. Additionally, the formation of forelimbs and hindlimbs begins with the expression of T-box transcription factor Tbx5 or Tbx4 in the restricted region of the lateral plate mesoderm, where the forelimbs and hindlimbs go on to be formed [39]. Understanding the molecular mechanism of genes regulating limb muscle development will help to reveal the intrinsic relationship between limb muscle development and its function.

Weighted correlation network analysis (WGCNA) can use module characteristic genes or hub genes within the modules to summarize these clusters and correlate modules with external sample traits so as to achieve the purpose of screening modules and genes related to traits [40]. For example, Ding et al. found significant differences in the gene regulatory network that regulates pigeon muscle development and growth and identified four significant modules and twenty hub genes that influence pigeon muscle development and growth based on WGCNA analysis [41]. Li et al. analyzed gene expression networks and functional modules in chicken breast muscle in different periods by using WGCNA. Tian et al. identified key modules and biomarkers in breast cancer based on WGCNA [42].

In this study, we used RNA-seq technology and combined it with WGCNA analysis to identify significant modules and hub genes in the development of the Chinese alligator’s fetal hindlimb. Ultimately, we identified four distinct modules and twenty hub genes that regulate the hindlimb development in the Chinese alligator at different developmental stages. These hub genes include genes that regulate the development of embryonic and hindlimbs (*GLI2*, *TBX18*, *GLI3*, *EPHB3*, *ACTA1*, and *MYOZ1*), genes that regulate the production and development of the nervous system (*UGT8*, *FAM151A*, and *UNC13A*), genes that regulate metabolism (*DHRS7C*, *CIC5*, and *SNCA*), and genes whose function is unclear (*P2RY12* and *IGSF12*). Numerous studies have shown that these genes play a significant role in hindlimb development. For example, *GLI2* and *GLI3* have redundant and context-dependent functions in skeletal muscle formation [43]. *TBX18* can regulate chondrogenesis formation during limb development in mice [44]. *EPHB3* is involved in regulating the formation of blood vessels and the molding of chick embryos [45]. *ACTA1* is a very highly conserved protein with a crucial role in force generation and muscle contraction [46]. *FAM151A* and *UNC13A* regulate the production and development of the nervous system during the hindlimb development [47,48,49]. *DHRS7C*, *CIC5*, and *SNCA* can regulate fat and carbohydrate metabolism during hindlimb development [50,51,52,53]. According to the expression level of each hub gene, the expression level of genes controlling cartilage growth and neural development decreased with the progression of development. However, this trend was reversed in the expression of genes that regulate metabolic capacity (Appendix A). This suggests that the development of hindlimbs in Chinese alligators is a process that focuses on tissue development in the early stage and metabolic ability development in the late stage. Finally, several hub genes, although their functions have not been reported in detail, are involved in important pathways regulating hindlimb development and may play an important role in hindlimb development.

After analyzing four significant modules and developmental periods, we found that similar to most amniotic animals [54,55], the hindlimb development of the Chinese alligator embryos goes through the development of the embryonic structure and hindlimb in the early stage (H29) and the development of immunity and metabolic capacity in the late stage (H46). In addition, the development of the hindlimb of the Chinese alligator is a complex process regulated by multiple pathways. Additionally, most of the signaling pathways in which the four significant modules are enriched related to hindlimb development. For example, the calcium signaling pathway is necessary for the growth of skeletal muscle hypertrophy [56] and can coordinate with the WNT signaling pathway to regulate limb muscle development [57]. In addition, adenosine monophosphate-activated kinase (AMPK) is considered to be the master switch of metabolism, being capable of regulating the level of energy metabolism in hindlimbs [58,59]. It is worth mentioning that there are more enrichment pathways (e.g., glycolysis/gluconeogenesis, neuroactive ligand–receptor interaction, and HIF–1 signaling pathway) associated with adaptation to low-oxygen environments than in many animals, such as pigeon, chick, and mouse [12,60,61,62]. And these pathways may be crucially important in the late Chinese alligator’s adaptation to low-oxygen water environments.

In this study, we screened four significant modules and twenty hub genes that regulate the hindlimb development of Chinese alligator embryos and preliminarily investigated the potential roles of these modules and genes in the hindlimb development process. This study presents differential gene expression and gene ontology results for the hindlimb of Chinese alligators, thus providing data support for the comparative evolution and functional genomics of crocodilians.

## 5. Conclusions

In this study, RNA-sequencing technology was used to analyze the gene expression of hindlimb in the Chinese alligator embryos at different stages (H29, H35, H41, and H46). WGCNA analysis was performed on 1675 selected DEGs. Finally, 4 modules and 20 hub genes, which were significantly related to the hindlimb development of the Chinese alligator embryos at different periods, were selected. Our study describes gene expression in the hindlimb of the Chinese alligator at different times. This will help to reveal the molecular mechanism regulating the development of the hindlimb of the Chinese alligator.

## Figures and Tables

**Figure 1 animals-13-03126-f001:**
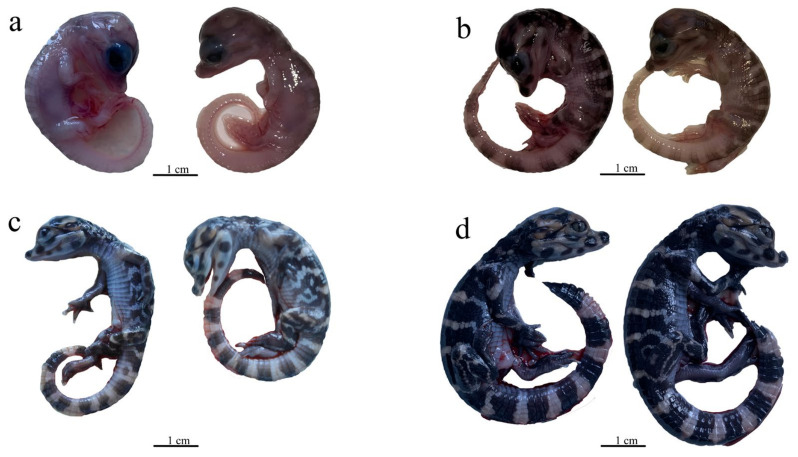
Embryo morphology of the Chinese alligator at four stages: (**a**) H29; (**b**) H35; (**c**) H41; (**d**) H46.

**Figure 2 animals-13-03126-f002:**
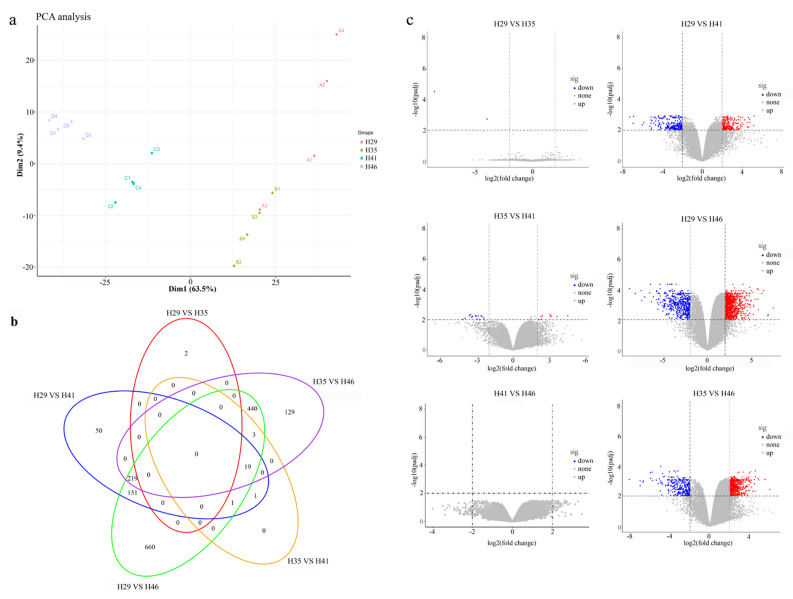
Analysis of gene expression patterns. (**a**) The results of PCA analysis among all samples. (**b**) The overlap results of DEGs for all time periods, and different colors represent different groups. (**c**) Volcano plot of differential gene expression analysis. Upregulated genes are marked in red, downregulated genes in blue, and nonsignificant genes in gray, respectively.

**Figure 3 animals-13-03126-f003:**
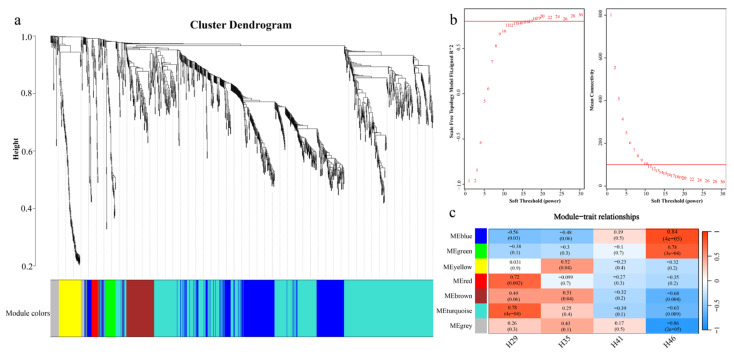
WGCNA analysis results of DEGs. (**a**) Clustering dendrogram of DEGs with dissimilarity based on the topological overlap; (**b**) soft threshold used in WGCNA analysis; (**c**) the correlation between modules and developmental stages.

**Figure 4 animals-13-03126-f004:**
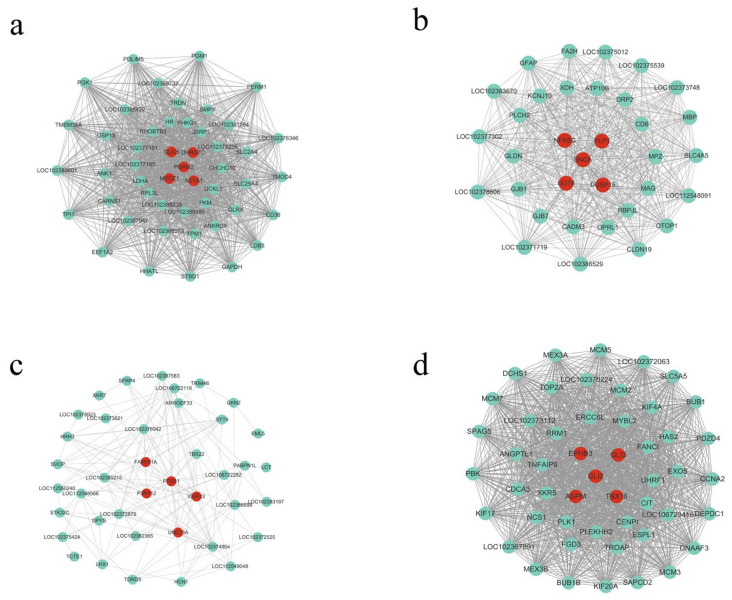
Visualization of the four modules related to development and growth of Chinese alligator muscle. (**a**–**d**) represent the blue, green, gray, and turquoise modules, respectively).

**Figure 5 animals-13-03126-f005:**
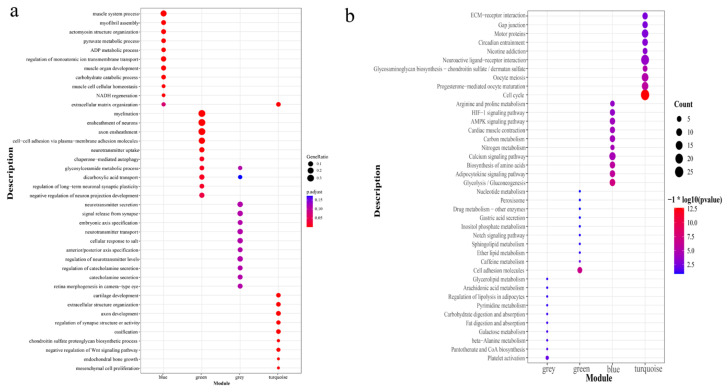
GO and KEGG enrichment results for DEGs in four significant modules (**a**,**b**) represent the GO and KEGG enrichment results, respectively).

**Figure 6 animals-13-03126-f006:**
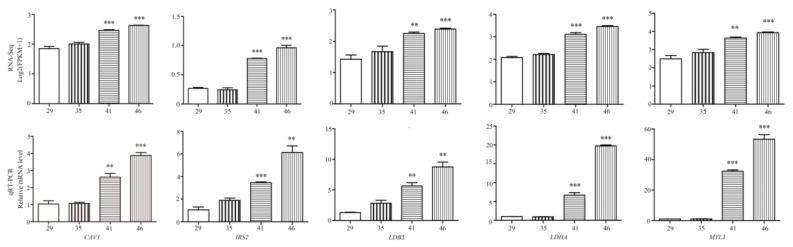
qRT-PCR validation of five differentially expressed genes in *Alligator sinensis*. (** *p* < 0.01, and *** *p* < 0.001).

**Table 1 animals-13-03126-t001:** Characteristics of limbs during embryonic stage of *Alligator sinensis*.

Period	Features
H29 (issue 20)	After this period, the differentiation of the toes began: the interdigit fissures between the first/second and second/third toes of the hindlimb reach the adult position, but the other fissures are not yet differentiated. The toe claw primordium begins to appear at the end of the first and second toes of the hindlimb.
H35 (issue 21)	Except for the fourth toe of the hindlimb, the other fingers (toes) form claws. The finger (toe) claw of the first, second, and third digits of the hindlimb form a small bend.
H41 (issue 24)	The claws of the fingers (toes) are more curved, and the differentiation of the scale tissue begins.
H46 (issue 25)	The pattern on the skin is clearly visible, and the differentiation of morphology is basically completed.

**Table 2 animals-13-03126-t002:** Data quality of *Alligator sinensis* transcriptome sequencing output.

Samples	Raw Reads	Raw Bases	Clean Reads	Clean Bases	Error Ratio (%)	Q20 (%)	Q30 (%)	GC Ratio (%)
A1	43,747,594	6.56 G	42,100,124	6.32 G	0.03	97.33	92.68	49.91
A2	41,711,106	6.26 G	40,232,798	6.03 G	0.03	97.45	92.96	49.73
A3	41,491,298	6.22 G	39,816,432	5.97 G	0.03	97.46	92.98	49.70
A4	42,042,466	6.31 G	40,533,992	6.08 G	0.03	97.60	93.24	49.79
B1	44,672,092	6.7 G	43,250,912	6.49 G	0.03	97.56	93.19	49.25
B2	40,775,384	6.12 G	39,288,900	5.89 G	0.03	97.56	93.16	50.14
B3	46,064,832	6.91 G	44,464,890	6.67 G	0.03	97.46	92.94	49.41
B4	46,125,062	6.92 G	44,434,414	6.67 G	0.03	97.70	93.48	49.81
C1	45,712,314	6.86 G	44,081,482	6.61 G	0.03	97.46	92.95	49.76
C2	46,326,252	6.95 G	44,523,064	6.68 G	0.03	97.60	93.30	50.64
C3	46,319,294	6.95 G	44,572,368	6.69 G	0.03	97.42	92.90	50.25
C4	45,632,232	6.84 G	44,076,128	6.61 G	0.03	97.35	92.74	50.28
D1	45,307,884	6.8 G	44,062,570	6.61 G	0.03	97.58	93.25	50.36
D2	44,863,392	6.73 G	43,390,542	6.51 G	0.03	97.54	93.16	50.80
D3	45,625,804	6.84 G	44,144,060	6.62 G	0.03	97.60	93.26	50.15
D4	46,945,240	7.04 G	45,648,164	6.85 G	0.03	97.34	92.52	50.53

## Data Availability

The datasets presented in this study can be found in online repositories. The name of the repository and accession number are as follows: NCBI; PRJNA941461.

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
