# Peer review of "Candidate Regulatory Genes for Hindlimb Development in the Embryos of the Chinese Alligator (Alligator sinensis)"

_animals, 2023, doi:10.3390/ani13193126_

Round 1
Reviewer 1 Report (New Reviewer)
Yang et al have carried out RNA sequencing of the Chinese Alligator hindlimb at different developmental stages. The authors provide very little results related to what genes they see expressed between the different stages and what significance these changes have in evolution/development. I believe this could be an interesting study if the authors focused on the specific gene changes in more detail over the developmental stages and how this relates to muscle development.
Comments:
There are many spelling mistakes/typos, I have listed a few below as examples that should be changed but the text should be further checked for errors:
Line 9: should “date support” be “data support”?
Line 13: RNA-seq sequencing, this should read RNA-sequencing.
Line 17: what is meant by the secondary animal? This isn’t clear.
Line 26: typo “tolal” should be “total”
Line 368: typo “please”
It would be good for the authors to detail what developmental events are occurring on 29, 35, 41 and 46 and why these days were chosen for analysis. It would be good for the authors to discuss the muscle anatomy of the alligator at these developmental stages and how they compare to other species e.g. human and mouse. This might raise some interesting findings if genes expressed are different.
It isn’t clear how the authors analysed just the “muscles”, there was no FACS/selection of the muscle cells. The sequencing was carried out on the entire hindlimb at each developmental stage and then during the analysis the muscle was selected based on gene expression? This should be made clearer in the text, simply saying development of hindlimb instead of hindlimb muscle would clear up this confusion.
Figure annotations are not legible in the version I have been sent so cannot read the results as the authors have intended. Unfortunately, there are no genes outlined in any of the figures representing the RNA sequencing data. It would be good to include a figure which shows genes upregulated between the different developmental stages.
It would be good for the authors to focus in on specific gene changes in the result section and how this can relate to alligator embryonic development, this has only been briefly touched upon in the discussion.
There are many typos/spelling mistakes that I have highlighted in my above comments that need to be addressed.
Author Response
- Yang et al have carried out RNA sequencing of the Chinese Alligator hindlimb at different developmental stages. The authors provide very little results related to what genes they see expressed between the different stages and what significance these changes have in evolution/development. I believe this could be an interesting study if the authors focused on the specific gene changes in more detail over the developmental stages and how this relates to muscle development.
- Thank you for your valuable comments, adding new ideas to our article analysis. Based on your comments, we increased the amount of hub gene expression analysis and we found that the expression level of genes controlling cartilage growth and neural development decreased with the progression of development, such as GLI2, TBX18, GLI3, EPHB3, MYOZ1, FAM151A, UNC13A. But this trend was reversed in the expression of genes (DHRS7C, CIC5, SNCA) that regulate metabolic capacity (Figure S1). This suggests that the development of hind limbs in Chinese alligators is a process that focuses on tissue development in the early stage and metabolic ability development in the late stage.
Comments:
There are many spelling mistakes/typos, I have listed a few below as examples that should be changed but the text should be further checked for errors:
- Line 9: should “date support” be “data support”?
- Thanks for your careful reading of our manuscript and pointing out our many spelling mistakes.This is our oversight. After correcting the spelling mistakes you have raised, we have carefully reviewed our manuscript to ensure that there are no more spelling mistakes in our manuscript, Please check in the modified version.
- Line 13: RNA-seq sequencing, this should read RNA-sequencing.
- Thank you for pointing out thiserror, And this error has been corrected in this vision.
- Line 17: what is meant by the secondary animal? This isn’t clear.
- Thank you for asking this question, which will help our manuscript to be widely read. Because our mistakes in the last version caused you confuse in reading, in this revision version, we have changed this part of the description to “Crocodilians, which are a kind of animal secondary adaptation to an aquatic environment...”
- Line 26: typo “tolal” should be “total”
- Thank you for pointing out thiserror for us, and this error has since been corrected, please check it in this revision version.
- Line 368: typo “please”
- Thanks for your careful reading of our manuscript and pointing out our many spelling mistakes.This is our oversight, and we are deeply sorry for the inconvenience caused to your reading. After correcting the spelling mistakes you have raised, we have carefully reviewed our manuscript to ensure that there are no more spelling mistakes in our manuscript, Please check in the modified version.
- It would be good for the authors to detail what developmental events are occurring on 29, 35, 41 and 46 and why these days were chosen for analysis. It would be good for the authors to discuss the muscle anatomy of the alligator at these developmental stages and how they compare to other species e.g. human and mouse. This might raise some interesting findings if genes expressed are different.
- Thank you for your important question. We divide the development period according to the morphological and developmental characteristics of the hind limbs of Chinese alligators, and the selected periods are representative. For example, day 29 is the time when the hind limbs of Chinese alligators begin to differentiate into toes, and day 35 is the time when the toes are basically differentiated and formed into claws. On day 41, the appendages begin to develop scales, after which the skin begins to show patterns. By the 46th day, the appendage development of the Chinese alligator was basically complete, and there was no obvious change in morphology after that. Please refer to the following references for the morphological characteristics and staging basis of fetal hind limbs of Chinese alligator:
Hua, T., et al., Morphogenesis of the Limb of Chinese Alligator. Journal of Anhui Normal University, 1995. 18(3): p. 38-41. (In Chinese)
Hua, T., C. Wang, and B. Chen, Stages of Embryonic Development for Alligator sinensis. Zoological Research, 2004. 25(3): p. 263-271.
Ferguson MMJ.1985.Post-laying stages of embryonic development for
crocodilians.In : Webb GJ W. Wildlife Management :Crocodiles and Alligators.London:Academic Press.427-444.
- In addition, we tried to compare the limb development process with that of other animals, but because of the different species of embryonic development dividing the period of the standard is not uniform, we could not draw a reliable conclusion. However, in alligators, the development process of hind limbs is relatively conservative, and the entire development stage can be divided into 28 stages, the specific staging standards can be referred to the above references
- It isn’t clear how the authors analysed just the “muscles”, there was no FACS/selection of the muscle cells. The sequencing was carried out on the entire hindlimb at each developmental stage and then during the analysis the muscle was selected based on gene expression? This should be made clearer in the text, simply saying development of hindlimb instead of hindlimb muscle would clear up this confusion.
- Thank you for your valuable suggestion. We sequenced the entire hind limb directly instead of FACS/selection of the muscle cells, which reflects not just the development of the muscle, but the development of the entire hind limb.We found that our study objectives are more relevant to the development of the hind limb than the development of hind limb muscles after careful consideration. Finally, we modify this statement in the paper, please check it.
- Figure annotations are not legible in the version I have been sent so cannot read the results as the authors have intended. Unfortunately, there are no genes outlined in any of the figures representing the RNA sequencing data. It would be good to include a figure which shows genes upregulated between the different developmental stages.
- Thanksfor your suggestion. This suggestion is of great scientific significance. We have modified the graphic annotations to be more specific and detailed. And the expression levels of each hub gene in 4 periods were added it this study.
- It would be good for the authors to focus in on specific gene changes in the result section and how this can relate to alligator embryonic development, this has only been briefly touched upon in the discussion.
- Thank you for your valuable suggestion. In this version, we increased the results of the amount of expression of each hub gene in 4 periods. And to correlate it with the development of the hind limbs of Chinese alligators, we found that the hindlimb development of the embryonic of the Chinese alligator goes through the development of embryonic structure, nervous system and muscle in the early stage. Thank you for your suggestions to enrich our research.
Reviewer 2 Report (New Reviewer)
The manuscript by Yang et al. describes a differential gene expression study in hind limbs of Chinese alligators. This paper would benefit from several improvements before it is published.
1. The manuscript describes the interesting situation of the hind limbs of alligators being more powerful than the fore limbs. However, the manuscript does not involve any experiments concerning forelimbs. Why do alligator hind limbs develop stronger than forelimbs? Not addressing this question in the manuscript is a missed opportunity.
2. The gene results presented here have long been known using other model organisms. What new insights are gained from the results presented here? Perhaps comparing the data sets presented here to those of other animals may provide context.
This manuscript reads as though it has passed through a translation algorithm. The sentence structure is distracting and hinders interpretation. Some examples:
Limb muscle development is the most important aspect of animal growth and development
so the development and differentiation of hind limb muscle is of more important significance for the survival of the Chinese alligator
Author Response
The manuscript by Yang et al. describes a differential gene expression study in hind limbs of Chinese alligators. This paper would benefit from several improvements before it is published.
- The manuscript describes the interesting situation of the hind limbs of alligators being more powerful than the fore limbs. However, the manuscript does not involve any experiments concerning forelimbs. Why do alligator hind limbs develop stronger than forelimbs? Not addressing this question in the manuscript is a missed opportunity.
- Thank you for your valuable comments. The conclusion that the hind limbs of Chinese alligator are stronger than the forelimbs is obtained from the later morphological observation. Since we only focused on the important role of hindlimb development in the survival of Chinese alligators at the beginning, we ignored the importance of comparing the differences in fore-hindlimb development during the experimental design, and finally lost such an important research opportunity.
- In addition, the purpose of this study is to reveal the molecular regulatory network of hind limb development in Chinese alligators. The original expression may cause confusion to you and readers, so we have revised this part of the expression to “For the Chinese alligator, the hindlimb is one of the main sources of power, and its development and differentiation will directly influence the survival ability in the wild. ”
- Finally, we are deeply aware that this problem you raised is a major regret for our research, so we will continue to start the development of forelimbs in the future
- The gene results presented here have long been known using other model organisms. What new insights are gained from the results presented here? Perhaps comparing the data sets presented here to those of other animals may provide context.
- Thank you for your advice. This question and suggestion have profound scientific significance. Based on your suggestion, we made a comparison of our analysis results with other species and we found more enrichment pathways (glycolysis / gluconeogenesis, neuroactive ligand−receptor interaction and HIF−1 signaling pathway) are associated with adaptation to low oxygen environments than in many animals, such as pigeon, chick and mouse. As a kind of animal secondary adaptation to an aquatic environment, the Chinese alligators can live in water for a long time, which is closely related to the adaptation to the low oxygen condition in water environment. And the hypoxia adaptability of Chinese alligator is closely related to these signaling pathways.
- On the other hand, the aim of this study is to screen the key genes that regulate the development of hind limbs of Chinese alligators. The specific functions of these genes in the development of hind limbs of Chinese alligators need further work. These further works will be more conducive to revealing the specific process of hind limb development of Chinese alligator, and also provide the background for limb development of other species.
Comments on the Quality of English Language
This manuscript reads as though it has passed through a translation algorithm. The sentence structure is distracting and hinders interpretation. Some examples:
Limb muscle development is the most important aspect of animal growth and development
so the development and differentiation of hind limb muscle is of more important significance for the survival of the Chinese alligator.
- Thank you for your comments. I apologize for the trouble caused by our language. For this reason, we have extensively edited the English language for our paper through MDPI company.
Reviewer 3 Report (New Reviewer)
The Chinese alligator is an endangered species unique to China, and its living habits, development processes, and other aspects are worth studying. This study mainly focuses on the gene expression patterns of the hindlimb muscles of Chinese alligators at different stages of embryonic development. Through RNA-seq, differentially expressed genes of the hindlimb muscles at different developmental stages were obtained. Using WGCNA analysis strategy, different gene expression modules and hub genes were identified, providing a deeper understanding of the development process of Chinese alligators.
Here are some comments for this manuscript:
1. In this manuscript, the embryonic developmental process was divided into four stages by the feature of hindlimb. Is the division of the embryonic stage of the Chinese alligator based on the features of its hindlimbs? If so, please provide relevant article citations. If not, is this division reasonable?
2. Where is the annotation of Figure 2?
3. In Figure 2A, A3 is clearly more similar in gene expression to group B, is it due to poor sample repeatability
4. Lines 235-236, ‘DEGs from the blue, gray, grey, and turquoise’ and line 240, ‘represent the blue, green, green, and turquoise’. Inconsistency before and after.
5. Lines 236-237, the texts described ‘…construct the gene co-expression network’ but what is marked in parentheses is ‘Figure 5’, which is GO and KEGG enrichment results.
6. Line 240, d should be capitalized.
7. Line 260, there is a space before the period.
8. Line 282, should not be capitalized after a comma.
Author Response
The Chinese alligator is an endangered species unique to China, and its living habits, development processes, and other aspects are worth studying. This study mainly focuses on the gene expression patterns of the hindlimb muscles of Chinese alligators at different stages of embryonic development. Through RNA-seq, differentially expressed genes of the hindlimb muscles at different developmental stages were obtained. Using WGCNA analysis strategy, different gene expression modules and hub genes were identified, providing a deeper understanding of the development process of Chinese alligators.
Here are some comments for this manuscript:
- In this manuscript, the embryonic developmental process was divided into four stages by the feature of hindlimb. Is the division of the embryonic stage of the Chinese alligator based on the features of its hindlimbs? If so, please provide relevant article citations. If not, is this division reasonable?
- Thank you for your important question. We divide the development period according to the morphological and developmental characteristics of the hind limbs of Chinese alligators, and the selected periods are representative. For example, day 29 is the time when the hind limbs of Chinese alligators begin to differentiate into toes, and day 35 is the time when the toes are basically differentiated and formed into claws. On day 41, the appendages begin to develop scales, after which the skin begins to show patterns. By the 46th day, the appendage development of the Chinese alligator was basically complete, and there was no obvious change in morphology after that. Please refer to the following references for the morphological characteristics and staging basis of fetal hind limbs of Chinese alligator:
- Hua, T., et al., Morphogenesis of the Limb of Chinese Alligator. Journal of Anhui Normal University, 1995. 18(3): p. 38-41. (In Chinese)
- Hua, T., C. Wang, and B. Chen, Stages of Embryonic Development for Alligator sinensis. Zoological Research, 2004. 25(3): p. 263-271.
- Where is the annotation of Figure 2?
- Thank you for your careful reading and timely pointing out our mistakes. This is an oversight on our part, and I apologize for any confusion these omissions may have caused you
- In Figure 2A, A3 is clearly more similar in gene expression to group B, is it due to poor sample repeatability
- Thank you for your timely clarification, this problem does exist.
- Lines 235-236, ‘DEGs from the blue, gray, grey, and turquoise’ and line 240, ‘represent the blue, green, green, and turquoise’. Inconsistency before and after.
- Thank you for your careful reading and timely pointing out our mistakes. This is an oversight on our part, and I apologize for any confusion these omissions may have caused you
- Lines 236-237, the texts described ‘…construct the gene co-expression network’ but what is marked in parentheses is ‘Figure 5’, which is GO and KEGG enrichment results.
- Thank you for your question, which is an oversight of ours. Here should be figure 4, which has been corrected
- Line 240, d should be capitalized.
- Thank you for pointing out our mistakes in time. In this version, we have uniformly replaced the letters in the numbers with lowercase letters. Please check it in the manuscript
- Line 260, there is a space before the period.
- Thank you for pointing out our mistakes in time. We have corrected the error and reviewed the entire manuscript to avoid a recurrence
- Line 282, should not be capitalized after a comma.
- Thank you for pointing out our mistakes in time. We have corrected the error and reviewed the entire manuscript to avoid a recurrence
This manuscript is a resubmission of an earlier submission. The following is a list of the peer review reports and author responses from that submission.
Round 1
Reviewer 1 Report
The authors try to use transcriptomics to identify the regulatory genes involved in the muscle development of the Chinese Alligator hindlimb. This manuscript is potentially important and interesting, but it suffers from poor analyses and organization. The authors may be unfamiliar with the transcriptomics, and substantial revisions are required. The detailed comments are listed below.
Major point
1 The experiment is designed to track the gene expression pattern with the development of the hind limb, and the PCA scatter plot in Figure 2 clearly indicates a gradual shift in gene expression pattern with the time or developmental stages. Therefore, the author should use more proper analyzing approach, rather than pairwise comparisons, to screen the differently expressed genes (DEGs), e.g., WGCNA, time-series analysis (R package Mfuzz), PLS models, as well as conducting Kruskal-Wallis test or one-way ANOVA on the gene expression tables. Only the methods which consider the fluctuations of gene expression levels across the group can present the variations of the transcriptome correctly.
2 The transcriptomic analyses are superficial. For example, the authors only concern the gene numbers and the category of the enriched items, while the variation trends of the KEGG pathways are not mentioned. How can these results reflect the molecular processes underlying the muscle development? More importantly, the author should highlight their main point and provide some novel insight into the muscle development of the reptiles, rather than give us a set of genes or KEGG items. For example, since the muscle is a major metabolic organ, the author should present us the variations of metabolic pathways with the development.
3 The authors emphasized the ‘regulatory’ in the title. However, they do not screen these genes de novo (e.g., based on the fold changes or p values of gene expression changes), but query the literatures. The real-time qPCR can also achieve this, and why you conduct transcriptomics? The major advantages of transcriptomics include 1) finding the largest changes in the gene expression pool (e.g., find the critical regulator or effector); 2) the genes from different but related pathways/biological processes can be verified mutually; 3) give a relatively complete story. None of these are achieved. If the transcriptomics fail to find the regulatory genes, the authors should introduce the selected candidate genes in the introduction.
The title It may be better to revise it as “Candidate Regulatory Genes for Hindlimb Muscle Development in the Embryos of Chinese Alligator (Alligator sinensis)”
Line 15 one of the most important
Line 18-19 I don’t understand the logic.
Line 19-21 In this study, the expression patterns of genes related to hindlimb development were investigated for developing Chinese alligator fetal at days 29, 35, 41, and 46 with transcriptomics.
Line 22-23 This sentence is meaningless, as these three categories cover all the genes.
Line 23-26 Rephrase this part to make it concise and clear.
Line 94-99 More detailed description on tissue sampling is required.
Line 97 represent what?
Line 112 Illumina Hiseq/Miseq? Which one? And the detailed type.
Line 170-171 significant differences in the sample transcription levels? The transcriptional level of the total genome or certain genes? The PCA scatter plot present the overall similarity in gene transcriptional profile, rather than level.
Line 200-205 What’s the mean of this part? This information does not make sense.
Line 207 ‘and the significant differences were small’ What does this mean?
Check the grammatical mistakes, as well specialized vocabulary of transcriptomics, thoroughly.
Author Response
First of all, thank you for your careful review of my manuscript and your suggestions of great significance. I changed the analysis of my article, and significant changes have been made to the article.
My response to your comment is as follows for your review, please check it.
1 The experiment is designed to track the gene expression pattern with the development of the hind limb, and the PCA scatter plot in Figure 2 clearly indicates a gradual shift in gene expression pattern with the time or developmental stages. Therefore, the author should use more proper analyzing approach, rather than pairwise comparisons, to screen the differently expressed genes (DEGs), e.g., WGCNA, time-series analysis (R package Mfuzz), PLS models, as well as conducting Kruskal-Wallis test or one-way ANOVA on the gene expression tables. Only the methods which consider the fluctuations of gene expression levels across the group can present the variations of the transcriptome correctly.
- Thank you for your suggestion. I am sorry for the trouble caused to you by the unscientific analysis method adopted before. The suggestions you put forward are very practical. For DEGs, we re-adopted WGCNA analysis. Please see the revised manuscript for specific results.
2 The transcriptomic analyses are superficial. For example, the authors only concern the gene numbers and the category of the enriched items, while the variation trends of the KEGG pathways are not mentioned. How can these results reflect the molecular processes underlying the muscle development? More importantly, the author should highlight their main point and provide some novel insight into the muscle development of the reptiles, rather than give us a set of genes or KEGG items. For example, since the muscle is a major metabolic organ, the author should present us the variations of metabolic pathways with the development.
- Thank you for your suggestion. Functional enrichment analysis was performed on DEGs identified by WGCNA analysis that significantly affected the development of embryonic hindlimb muscles in Chinese alligator at different stages.And GO enrichment results showed that the DEGs in brown modules were enriched to 64 biological processes, such as connective tissue development, cartilage development and axon development (Figure 6a). DEGs in grey modules were not enriched to biological processes . DEGs in yellow modules were enriched to 30 biological processes, such as activation of immune response, regulation of immune effector process and positive regulation of cell projection organization (Figure 6c). DEGs in red modules were enriched to 12 biological processes, such as gliogenesis, adult behavior and locomotory behavior (Figure 6c). The significantly enriched GO molecular function and cellular component terms are listed in Table S2. The KEGG enrichment results showed that DEGs in the brown module were significantly enriched into the signalling pathways such as PI3K−Akt signaling pathway, human papillomavirus infection and neuroactive ligand−receptor interaction (Figure 7a). DEGs in the grey module were significantly enriched into the signalling pathways such as axon guidance, adrenergic signaling in cardiomyocytes and ECM−receptor interaction (Figure 7b). DEGs in the yellow module were significantly enriched into the signalling pathways such as omplement and coagulation cascades, neuroactive ligand−receptor interaction and csignaling pathway (Figure 7c). And DEGs in the red module were significantly enriched into the signalling pathways such as neuroactive ligand−receptor interaction, cAMP signaling pathway and Cell adhesion molecules (Figure 7d). For details, please see the revised manuscript.
3 The authors emphasized the ‘regulatory’ in the title. However, they do not screen these genes de novo (e.g., based on the fold changes or p values of gene expression changes), but query the literatures. The real-time qPCR can also achieve this, and why you conduct transcriptomics? The major advantages of transcriptomics include 1) finding the largest changes in the gene expression pool (e.g., find the critical regulator or effector); 2) the genes from different but related pathways/biological processes can be verified mutually; 3) give a relatively complete story. None of these are achieved. If the transcriptomics fail to find the regulatory genes, the authors should introduce the selected candidate genes in the introduction.
- Thank you for your suggestion. We have modified the title of the paper according to your suggestions, and modified the screening criteria for key genes based on gene connectivity in WGCNA analysis. Please check it in the article.
4 Line 18-19 I don’t understand the logic.
- Thank you for your suggestion. We apologize for the confusion and have reorganized the logic of this sentence. The modified version is “In the Chinese alligator, the hindlimb muscles are one of its main sources of power, so the development and differentiation of hind limb muscle is important for the survival of the Chinese alligator. Furthermore, a better understanding of molecular mechanism regulating the development of muscle in the Chinese alligator hindlimb, will provide date support of molecular phylogeny for the Chinese alligator conservation.”
5 In this study, the expression patterns of genes related to hindlimb development were investigated for developing Chinese alligator fetal at days 29, 35, 41, and 46 with transcriptomics.
- Thank you for your suggestion.It has been modified according to your suggestion, please refer to the modified version.
6 Line 22-23 This sentence is meaningless, as these three categories cover all the genes.
- Thank you for your suggestion.This sentence has been redescribed, please refer to the modified version.
7 Line 23-26 Rephrase this part to make it concise and clear.
- Thank you for your suggestion.This sentence has been redescribed, please refer to the modified version.
8 Line 94-99 More detailed description on tissue sampling is required.
- Thank you for your suggestion.And we have added a description of the sampling details, please refer to the modified version.
9 Line 97 represent what?
- Thank you for your question.We want to determine the different stages of hind limb development of Chinese alligator based on the anatomical results of earlier Chinese alligator and morphological data. Thus avoiding duplication of sampling periods.
10 Illumina Hiseq/Miseq? Which one? And the detailed type.
- Thank you for your suggestion. This was our mistake, and it has now been corrected. Please check it in the article.
11 Line 170-171 significant differences in the sample transcription levels? The transcriptional level of the total genome or certain genes? The PCA scatter plot present the overall similarity in gene transcriptional profile, rather than level.
- Thank you for your suggestion. This was our mistake, and it has now been corrected. Please check it in the article.
12 Line 200-205 What’s the mean of this part? This information does not make sense.
- Thank you for your suggestion. As the data has been added to the analytical methods, this section of content has awakened a redescription. Please check it in the article.
13 Line 207 ‘and the significant differences were small’ What does this mean?
- Thank you for your suggestion. As the data has been added to the analytical methods, this section of content has awakened a redescription. Please check it in the article.
Finally, I would like to thank you again for your valuable comments on my manuscript, which have provided me with new ideas for the mining and analysis of manuscript data.
Wish you well.
Liuyang Yang
Reviewer 2 Report
In this paper, Yang performed RNA-seq analysis to identify unique gene expression of the hindlimb muscular in Alligator. They found 7 candidate genes, CA3, CTNNB1, LEF1, MSTN, MYOD1, SFRP2, and TCFL2. Finally, they discussed about relationship these genes and their function to form muscle.
Overall, I could easily understand the results and what author did in this paper. However, the aim of this study was not clearly written. In the first Simple Summary, it is written that crocodiles have stronger muscles in their hind limbs than their forelimbs. From the point of view of current paper structure, I could not understand why the gene expression in crocodile muscle development over time was investigated. Reading this paper made me think that the authors would like to find out genes that induce the unique development of hindlimb muscles in crocodiles. To do so, authors should look for DEGs in comparison with, for example, forelimb and hindlimb RNA-seq data or hindlimb RNA-seq data from different species. Thus, this manuscript was not organized well regarding scientific purpose. I would like to ask authors to clarify the purpose and significance of their research. This paper seems to be just looking at gene expression in crocodile muscle development over time and no advance of the field of developmental biology. For these reasons, this paper should be revised.
Author Response
First of all, Thank you for your valuable comments on our manuscript. And my specific recovery is as follows:
- Overall, I could easily understand the results and what author did in this paper. However, the aim of this study was not clearly written. In the first Simple Summary, it is written that crocodiles have stronger muscles in their hind limbs than their forelimbs. From the point of view of current paper structure, I could not understand why the gene expression in crocodile muscle development over time was investigated. Reading this paper made me think that the authors would like to find out genes that induce the unique development of hindlimb muscles in crocodiles. To do so, authors should look for DEGs in comparison with, for example, forelimb and hindlimb RNA-seq data or hindlimb RNA-seq data from different species. Thus, this manuscript was not organized well regarding scientific purpose. I would like to ask authors to clarify the purpose and significance of their research. This paper seems to be just looking at gene expression in crocodile muscle development over time and no advance of the field of developmental biology. For these reasons, this paper should be revised.
- Thank you for your suggestion. The purpose and significance of the paper have been restated according to your comments. Please check it in the article.
Wish you well.